# Pollution and Health Risk Assessment of Potentially Toxic Elements in Groundwater in the Kǒnqi River Basin (NW China)

**DOI:** 10.3390/toxics12070474

**Published:** 2024-06-29

**Authors:** Yonglong Hu, Mamattursun Eziz, Liling Wang, Xayida Subi

**Affiliations:** 1College of Geographical Science and Tourism, Xinjiang Normal University, Urumqi 830054, China; a18048617279@163.com (Y.H.); wll18152166428@163.com (L.W.); xayida104@126.com (X.S.); 2Laboratory of Arid Zone Lake Environment and Resources, Xinjiang Normal University, Urumqi 830054, China

**Keywords:** groundwater, PTEs, pollution, health risk, Monte Carlo simulation

## Abstract

Potentially toxic elements (PTEs) pose a significant threat to the groundwater system and human health. Pollution and the potential risks of PTEs in groundwater in the Kǒnqi River Basin (KRB) of the northwest arid zones of China are still unknown. A total of 53 groundwater samples containing eight PTEs (Al, As, Cd, Cu, Mn, Pb, Se, and Zn) were collected from the KRB, and the pollution levels and probabilistic health risks caused by PTEs were assessed based on the Nemerow Index (NI) method and the health risk assessment model. The results revealed that the mean contents of Al, As, and Mn in the groundwater surpassed the Class III threshold of the Standard for Groundwater Quality of China. The overall pollution levels of the investigated PTEs in the groundwater fall into the moderate pollution level. The spatial distributions of contents and pollution levels of different PTEs in the groundwater were different. Health risk assessment indicated that all the investigated PTEs in groundwater in the KRB may pose a probabilistic non-carcinogenic health risk for both adults and children. Moreover, As may pose a non-carcinogenic health risk, whereas the non-carcinogenic health risk posed by the other seven PTEs in groundwater will not have the non-carcinogenic risks. Furthermore, As falls into the low carcinogenic risk level, whereas Cd falls into the very low carcinogenic risk level. Overall, As was confirmed as the dominant pollution factor and health risk factor of groundwater in the KRB. Results of this study provide the scientific basis needed for the prevention and control of PTE pollution in groundwater.

## 1. Introduction

Groundwater is one of the most critical sources of freshwater supply worldwide. Nearly two-thirds of the world’s population relies on groundwater for survival and daily activities [1]. In China, nearly 70% of daily water supply and 40% of irrigation water rely on groundwater [2]. In arid regions, groundwater is a vital component of water ecosystems and is of great significance to ecosystem stability, human health, and agricultural production [3]. Groundwater is even more critical as it plays a crucial role in regional water supply and the eco-environmental security of arid regions [4,5].

Groundwater pollution has attracted interest worldwide for its determinant effects on water supply and food security [6,7]. Potentially toxic elements are the most common pollutant in the groundwater system, which has been reported to have different degrees of pollution around the world [8,9]. In recent years, with the rapid development of modern agriculture and industry, pollution of groundwater by PTEs has grown to be a complex environmental problem due to their high toxicity, non-degradability, and persistence [10,11]. Toxic elements from industry, agriculture, and natural sources can pollute groundwater through wastewater discharge and infiltration [12,13]. PTEs that accumulate in groundwater can directly or indirectly threaten human health via direct ingestion and dermal absorption [14,15], and can also threaten natural ecosystems through water recycling [16]. Among them, As, Cd, and Pb are extremely toxic to the human body and animals, even at low content levels [17]. Other PTEs, such as Al, Cu, Mn, Se, and Zn, are necessary for the metabolism, but they can become toxic when their content in groundwater exceeds the tolerance limit of the human body [2]. For example, consumption of groundwater containing high contents of As and Cd, are known to be carcinogenic, can affect the functions of the kidneys, and can cause liver, lung, and bladder cancers [18,19]. Extended exposure to Zn at high levels increases the risk of developing neurological and behavioral disorders [8]. Lead exposure can cause kidney and brain damage, cognitive decline, and cardiovascular diseases [20]. Copper exposure can also cause liver damage and impaired kidney function [15]. Elevated aluminum manganese and selenium levels influence the brain and liver, and cause behavioral problems [20]. In addition, PTEs can also affect the physical and chemical properties of groundwater and pose significant challenges to the sustainable use of groundwater [21]. PTEs discharged into the groundwater system are mainly caused by natural origins (soil-water interactions, geochemical background, or mineral leaching) and anthropogenic activities (agricultural pollution, industrial pollution, or wastewater discharge) that lead to the pollution of groundwater systems [22]. Moreover, the use of polluted groundwater for agricultural irrigation may lead to the accumulation of PTEs in soil, and ultimately result in reduced crop yields and harm human health [23]. Therefore, the health risks of toxic elements in groundwater have become a critical research concern worldwide [24].

Health risk assessment of PTEs in groundwater is of high importance due to its direct relation to human life [6,25,26]. Therefore, periodical monitoring of PTEs in groundwater is required to identify their potential risks. Pollution assessment of PTEs in groundwater is necessary before estimating potential health risks. The Nemerow index (NPI) [27], heavy metal pollution index (HPI) [28], and water quality index (WQI) [29] are effective methods for assessing pollution levels of PTEs in groundwater. However, these traditional methods have certain limitations. They require a clear critical limit value without considering factors such as ingestion rate, body weight, exposure duration, and exposure frequency [30]. The health risk assessment model introduced by the US Environmental Protection Agency utilizes clear values and assumptions to estimate health risks and considers related parameters, thus can provide a more comprehensive evaluation of potential health risks [15]. Recently, health risk assessment based on the Monte Carlo Simulation is considered a more powerful technique because it considers the variability and uncertainty in input parameters, decreases uncertainty related to exposure parameters, and thus enhances the accuracy of potential health risk assessments [31,32,33].

In China, nearly 80% of shallow groundwater has been polluted to varying degrees [14]. Among such, pollution of groundwater by PTEs should not be ignored. Recent reports [20,34,35] indicated that various anthropogenic activities and low groundwater recharge have led to the pollution and degradation of the quality of many groundwater systems in the northwestern arid zones of China. So far, however, the pollution and health risks of PTEs in the groundwater of the Kǒnqi River Basin (KRB), which is a major pear production area in China and also one of the main industrial areas in Xinjiang in northwest China, are nearly unknown. It should be noted that, under the influence of anthropogenic activities such as industry and rapid expansion of agriculture, the various PTEs of water cycle in the KRB have drastically changed and caused a series of water problems, such as groundwater pollution and the decline of the groundwater table [36,37].

The main goals of the present study are (1) to determine the contents and spatial distributions of eight PTEs, including Al, As, Cd, Cu, Mn, Pb, Se, and Zn in the groundwater of the KRB in the northwest arid regions of China, (2) to quantify the pollution levels of PTEs in groundwater, and (3) to assess the potential health risks of PTEs in groundwater, based on the US EPA health risk assessment model. The results of this study will provide guidance for groundwater pollution control and management in the KRB, as well as other similar aquatic ecosystems in arid regions.

## 2. Materials and Methods

### 2.1. Study Area

The Kǒnqi River Basin (85°40′ E~86°20′ E, 41°20′ N~41°40′ N) is located in Xinjiang in the northwestern arid regions of China (Figure 1). It is located on the southern slopes of the Tianshan Mountains and in the northern parts of the Taklimakan Desert. The KRB is positioned in the central part of the Eurasian continent and experiences an arid and hot summer climate, as well as cold winters, which is characteristic of a typical continental arid desert climate with an annual mean precipitation of 11.4 mm, evaporation of 2800 mm, and temperature of 10.5 °C.

The groundwater in the KRB belongs to the unconsolidated–layer–pore water type, primarily found in the alluvial–pluvial layer, with abundant occurrences of phreatic and confined water in a layered structure. The topographical features consist of vast mountain ranges, oases on narrow plains, and expansive deserts. The terrain is high in the north and low in the south. Agricultural irrigation in the KRB requires the extraction of groundwater. The KRB is rich in mineral resources, such as coal, oil, iron, and manganese. A new generation of petroleum and petrochemical industry has been forming in the KRB in recent decades [38]. The soil parent material has a high salt content, and the secondary salinization of riverbank soil is severe. The main soil types are brown desert soil and salinized soil, and the main crops are cotton and pepper.

### 2.2. Groundwater Sampling

Based on the Chinese national standard HJ/T 164–2004 [39], a total of 53 groundwater samples were collected from the KRB in August 2023, with a period of no significant precipitation within two weeks. The locations of sample sites are also depicted in Figure 1. Groundwater samples for daily use and irrigation water were collected from pumping wells or wells. To ensure proper sample preservation, the groundwater samples were stored in polypropylene bottles that had been pre-washed with deionized water. After the field survey sampling, the samples were immediately transported to the laboratory. Before the laboratory analysis, all samples were filtered using disposable syringes (10 mL, filter pore size 0.45 μm) then acidified with nitric acid to maintain a pH below 2. The samples were stored in a controlled environment at 4 °C before laboratory analysis.

### 2.3. Chemical Analysis

The collected groundwater samples were entrusted to the “Xinjiang Shuiqingqing Environmental Monitoring Technology Service Co., Ltd.” to determine the contents of PTEs. The contents of Al, As, Cd, Cu, Mn, Pb, Se, and Zn were determined, per the Chinese National Standard detailed in HJ 700–2004 [40], using an inductively coupled plasma–mass spectrometer (ICP–MS, Perkin Elmer, Waltham, MA, USA) (RSD < 5%). The detection limits for Al, As, Cd, Cu, Mn, Pb, Se, and Zn were 1.15, 0.12, 0.05, 0.08, 0.12, 0.09, 0.41, and 0.67 µg/L, respectively.

### 2.4. Quality Assurance and Quality Control (QA/QC)

All groundwater samples, laboratory blanks and standard spiked samples were analyzed for QA/QC. Results of blank tests indicated that their measured values were lower than the detection limits. The average recovery rate for all target PTEs ranged from 94.38% to 108.8%. To ensure the measured data quality, calibration curves were produced and the contents of PTEs in each sample were tested three times. The relative standard deviation (RSD) for the test substances was maintained below 15%, which complies with the requirements of the US EPA (RSD < 30%) [3]. QA/QC ensured the accuracy and reliability of the chemical analysis results.

### 2.5. Pollution Level of PTEs in Groundwater

The Nemerow index (NI) method [27] is used for comprehensively assessing both single and integrated pollution levels of PTEs in groundwater. It is calculated as follows:

*I_i_* = *M_i_*/*S*_i_(1)
(2)$$\mathrm{NI}=\sqrt{\left({I_{max}}^{2}+{I_{mean}}^{2}\right)/2}$$
where *I_i_* represents the pollution level of a single element *i*, and NI represents the overall groundwater quality of PTEs [2]. *M_i_* represents the content of element *i* in the surveyed groundwater samples, *S_i_* is the evaluation standard of element *i*. The *I_max_* and *I_mean_* are the maximum and mean values of *I_i_*, respectively. The Class III threshold of the Standard for Groundwater Quality, suggested by the AQSIQ [40], was selected as the evaluation standard. The classification standards [2] for the pollution degree of *I_i_* and NI are given in Table 1.

### 2.6. Probabilistic Health Risk Assessment

In this study, we adopted the health risk assessment model [41] to evaluate the probabilistic health risks of exposure to PTEs through oral ingestion and dermal contact. We specifically considered children as a sensitive group, while adults are generally regarded as the general population. Traditional human health risk assessments have primarily relied on fixed parameters to calculate associated risks [42]. Uncertainties in risk characterization parameters may lead to biases in risk assessment [31,43]. Therefore, probabilistic methods based on the Monte Carlo Simulation (MCS) were applied in this study. The MSC involve subjecting parameters to specific ranges, allow for a more accurate risk assessment, and demonstrate the impact of different parameters on risk assessment outcomes [44,45]. MCS obtain the probability distribution of health risks of groundwater PTEs by incorporating a large number of random samples consistent with certain probability distributions into the mathematical model, with a random simulation iteration count of 10,000 [45,46]. Based on existing research [47], Al, As, Cd, Cu, Mn, Pb, Se, and Zn were categorized as non-carcinogenic PTEs, while As and Cd were listed as both non-carcinogenic TEs and carcinogenic PTEs.

The exposure to PTEs in the groundwater was characterized by the chronic daily intake (CDI, mg/kg/d) introduced by the US EPA [41]. The formula for calculating the CDI of the *j*th element at the *i*th sample site is as follows [48]:(3)$$\mathrm{CD}I_{ij\;oral}=(C_{ij}\times IR\times EF\times ED)/(BW\times AT)$$
(4)$$\mathrm{CD}I_{ij\;derm}=(C_{ij}\times SA\times PC\times ET\times EF\times ED\times CF)/(BW\times AT)$$
where $C_{ij}$ is the *j*th element at the *i*th sample site (μg/L); $\mathrm{CD}I_{ij\;oral}$ is the CDI from the ingestion route of the *j*th element at the *i*th sample site; $\mathrm{CD}I_{ij\;derm}$ is the CDI from the dermal contact route of the *j*th element at the *i*th sample site; *IR* is the ingestion rate (1.8 L/d for adults, and 0.70 L/d for children) [49]; *EF* is the exposure frequency (350 (180, 365) days/year for both adults and children) [31]; *ED* is the exposure duration (24 years for adults, and 6 years for children) [50]; *BW* is the body weight (70.0 kg for adults, and 21.2 kg for children) [46]; *AT* is the averaging time (for non-carcinogenic risk, 8760 days for adults, and 2190 days for children; for carcinogenic risk, 25,550 days for both adults and children) [31]; *SA* is the skin surface area (16,600 cm^2^ for adults, and 8000 cm^2^ for children) [46]; *PC* is the absorption coefficient of the human body (1 × 10^−3^ for As, Cd, Cu, Mn, Pb, Al, and Se, and 6 × 10^−4^ for Zn) [50]; *ET* is the event duration (0.20 (0.13, 0.33) hour/event), and *CF* is the conversion factor (0.001 L/cm^3^) [50].

Then, the potential non-carcinogenic health risk (HI) of PTEs in the groundwater was calculated as the hazard quotient (HQ) as follows:(5)HQ=CDI/RfD
(6)HI=ΣHI
where *RfD* is the reference dose for the ingestion and dermal contact routes. The *RfD* is given in Table 2 [48,51]. Depending on the risk classification criteria [52], when HI < 1, PTEs in groundwater will not have non-carcinogenic risks. HI > 1 indicates a probabilistic non-carcinogenic health risk.

The probabilistic carcinogenic health risk of PTEs in the groundwater was calculated as the carcinogenic risk index (CR) as follows:(7)CR=CDI×SF
(8)TCR=ΣCR
where *SF* is the reference dose for the ingestion and dermal contact routes, TCR is the total carcinogenic risk index of all the PTEs. The values of *RfD* and *SF* are given in Table 2 [15,51].

The classification of the carcinogenic risk degree of CR and TCR are as follows: very low carcinogenic risk (CR or TCR < 1 × 10^−6^), low carcinogenic risk (1 × 10^−6^ ≤ CR or TCR < 1 × 10^−4^), moderate carcinogenic risk (1 × 10^−4^ ≤ CR or TCR < 1 × 10^−3^), high carcinogenic risk (1 × 10^−3^ ≤ CR or TCR < 0.1), and extremely high carcinogenic risk (CR or TCR > 0.1) [53,54].

### 2.7. Monte Carlo Simulation (MCS)

The MCS is a statistical mathematical theory adapted to uncertainty analyses in potential health risk assessment. To minimize uncertainty and provide a more robust health risk assessment of PTEs in the groundwater for the residents, the MCS approach was employed in this study to simulate the uncertainty of exposure parameters used for health risk assessment calculations associated with PTEs in the groundwater [7,55].

### 2.8. Statistical Analysis

For the statistical analysis of groundwater data, the IBM Statistical Package for the Social Sciences (SPSS) 25.0 (Chicago, IL, USA) was used. A normal distribution test was conducted of the variables, and then an abnormal distribution test was used to analyze the PTE contents after logarithmic transformation. The GIS-based Ordinary Kriging interpolation method, which has been widely used in geostatistics [2], was applied in order to map the spatial distribution of PTEs in the groundwater. The ArcGIS 10.2 software (Esri, CA, USA) was used for mapping the spatial distribution of PTEs. Sensitivity analysis was conducted using the MATLAB 2021 software(MathWorks, NM, USA).

## 3. Results

### 3.1. The Contents of PTEs in Groundwater

Table 3 presents a basic statistical summary of the investigated PTEs in groundwater in the KRB, and the Class III threshold of the Standard for Groundwater Quality of China (GB/T 14848—2017) [40]. As shown there, the mean contents of Al, As, Cd, Cu, Mn, Pb, Se, and Zn in groundwater in the KRB are 233 μg/L, 21 μg/L, 0.35 μg/L, 32 μg/L, 180 μg/L, 6 μg/L, 7 μg/L, and 156 μg/L, respectively. The mean contents of Al, As, and Mn in the groundwater surpass the Class III thresholds by factors of 1.17, 2.10, and 1.80 times, respectively. The mean contents of the other five PTEs are lower than the corresponding threshold values. The maximum contents of Al, As, and Mn in the groundwater surpass the corresponding threshold values by factors of 4.99, 17.60, and 10.47 times. It should be pointed out that the maximum contents of Cd, Pb, and Se also surpass the corresponding threshold values by factors of 1.10, 4.59, and 19.73 times, respectively.

However, except for Cu and Zn, a considerable percentage of groundwater samples exceed the Class III threshold of the National Standard for Al (49.06%), As (35.85%), Cd (1.87%), Mn (45.28%), Pb (20.75%), and Se (15.09%). According to the above analysis, it appears that Al, As, Mn, Pb, and Se are particularly more abundant in groundwater in the KRB. The calculated coefficient of variations (CV) of all PTEs have values exceeding 100%, with strong variability (CV > 50%), indicating a significant level of spatial variability [2]. The skewness and kurtosis of these PTEs are also relatively high. This indicates that there are noticeable variations in PTEs contents among different sampling sites, which could potentially suggest pollution from specific sources [56,57]. However, the PTEs falling into the strong variability classes are likely to be influenced by anthropogenic factors.

### 3.2. The Spatial Distribution of PTEs in Groundwater

A GIS-based Ordinary Kriging (OK) interpolation was used for mapping the spatial distribution patterns of the contents of eight PTEs in groundwater in the KRB (Figure 2).

As shown here, the spatial distributions of the contents of different PTEs in the groundwater were different, with the distribution of the high-PTE-content areas in the groundwater showing a zonal distribution pattern, exhibiting characteristics of natural source input, as reported by Muyassar et al. [2]. In the case of Al, the high-content areas were primarily observed around Korla city, while the low-content areas for Al were observed around the southern parts of the KRB. The spatial distributions of As, Se, and Mn contents in groundwater in the KRB were relatively similar. The high-content areas of these three PTEs were mainly detected in the southwestern parts, while low-content areas of these three PTEs were primarily observed around western and northeastern parts in the KRB. The spatial distributions of Cu and Cd in groundwater were very similar, with the high-content areas of these two PTEs mainly distributed in the eastern parts, while low-content areas were observed around the northern and southern parts of the KRB. The spatial distributions of Pb and Zn contents were also relatively similar. The high-content areas of these two PTEs were mainly detected in the southeastern and southwestern parts, while low-content areas of them were primarily observed around the northeastern parts of the KRB.

Overall, the zonal distribution patterns of the PTE contents of groundwater in the KRB indicated that the PTEs originated mainly from natural sources. However, the PTEs in groundwater were affected by the leaching of PTEs from the surface, and the geochemistry of this basin also affects the presence of PTEs in groundwater [25].

### 3.3. Pollution Levels of PTEs in Groundwater

Two indices, *I_i_* and NI, were used to evaluate the pollution levels of PTEs in groundwater in the KRB (Table 4). As given in Table 4, the mean *I_i_* values of the investigated PTEs in groundwater samples in the KRB can be ranked as: As (2.107), Mn (1.804), Al (1.165), Se (0.744), Pb (0.646), Zn (0.156), Cd (0.071), Cu (0.032), with higher *I_i_* values representing higher degrees of pollution. According to the classification standard of pollution degree of *I_i_*, As in the groundwater falls into the moderate pollution level, while Al and Mn fall into the low pollution level. Se falls into the slight pollution level, and the other four PTEs fall into the no pollution level. However, the maximum *I_i_* values of Al, As, Mn, Pb, and Se in groundwater fall into the high pollution level, while the maximum *I_i_* values of Cd and Zn fall into the slight pollution level. The NI values of the investigated PTEs in groundwater in the Kǒnqi River Basin were in the range of 0.280–12.600, with a mean value of 2.750 at the moderate pollution level. Overall, the pollution levels of PTEs in groundwater in the KRB were relatively high. The CV values for *I_i_* and NI values of PTEs showed strong variability. This indicates that the pollution levels of each PTE in each groundwater sample have higher spatial variability, with point source pollution. However, As, among these eight PTEs, was the main pollution factor in groundwater in the KRB.

The spatial distribution patterns of the *I_i_* and NI values are shown in Figure 3. As illustrated in Figure 3, the spatial distribution patterns of *I_i_* of Al, Mn, and Pb show a “dotted-distribution” pattern, exhibiting a characteristic of possible point source pollution, according to the research results of Muyassar et al. [2]. The spatial distribution patterns of *I_i_* of As and NI values of PTEs in groundwater in the KRB were quite similar, with the high pollution areas of *I_i_* of As and NI values mainly distributed in the southeastern and southwestern parts. Moreover, the high As pollution area of groundwater was the biggest. This proves that As is the dominant pollution factor in groundwater in the KRB. The spatial distribution patterns of *I_i_* of Se showed a zonal-distribution pattern, *I_i_* of Se gradually decreased from the southwestern parts to other parts of the study area. The spatial distribution patterns of *I_i_* of Cd and Zn also showed a “dotted-distribution” pattern, while *I_i_* of Cu showed no pollution for the whole study area.

However, high-As groundwater is a serious environmental problem worldwide. As is a carcinogenic HM with high toxicity and listed in the “List of Toxic and Harmful Water Pollutants” by the Ministry of Ecology and Environment of China [2]. The KRB of Xinjiang is one of the high-As groundwater areas in the northwestern arid regions of China. Therefore, special attention should be paid to the higher pollution risks of As in groundwater in the KRB, considering its high toxicity and higher levels in this region.

### 3.4. The Non-Carcinogenic Health Risks of PTEs in Groundwater

Based on the Monte Carlo Simulation, the probability distributions of the non-carcinogenic risk (HI) for two groups (adults and children) under the oral ingestion and dermal contact exposure pathways of PTEs in groundwater in the KRB were obtained (Figure 4). The mean hazard quotient (HQ) values of PTEs for children and adults were found in the following order: As > Se > Pb > Mn > Al > Cu > Cd > Zn. This indicates that the non-carcinogenic risks of As and Se are higher than those of other PTEs in groundwater in the KRB. The mean HQ values of As for adults and children were 1.02 and 1.99, respectively. The HQ of children is higher than that of adults, which might be due to their lower weight, resulting in a relatively higher average daily exposure and a higher sensitivity to the external environment than adults [58]. A previous study [59,60] confirmed that the HI value of PTEs is generally higher for children than that for adults. The HQ values of the remaining seven PTEs in groundwater were less than 1 at a 95% confidence level. A related study [61] suggested that the non-carcinogenic health risk posed by PTEs in groundwater can be negligible, if the HQ value is below the acceptable risk threshold (HQ = 1) at a 95% probability. In this study, As reached the acceptable risk threshold at the cumulative probability of 76.78% and 61.74% for adults and children, respectively.

However, at a 95% probability, the HQ values of Al, As, Cd, Cu, Mn, Pb, Se, and Zn in groundwater were below the acceptable risk threshold, for both adults and children. This indicates that Al, Cd, Cu, Mn, Pb, Se, and Zn in groundwater in the KRB are not risky at a 95% confidence level. The above analysis proves that As in groundwater in the KRB may pose a probabilistic non-carcinogenic health risk, while the probabilistic non-carcinogenic health risks associated with the other seven PTEs were negligible. Therefore, As can be selected as the main non-carcinogenic health risk factor of groundwater in the KRB.

In terms of the overall non-carcinogenic health risk (HI) (Figure 5), the HI values for adults and children surpassed the acceptable risk threshold (HI = 1) at a 95% probability. This indicates that the investigated PTEs in groundwater in the KRB may pose a probabilistic non-carcinogenic health risk for both adults and children. Moreover, the mean HI value for children (2.54) was higher than that for adults (1.30); children face higher non-carcinogenic health risks than adults. This indicates that children have a higher cumulative non-carcinogenic health risk compared to adults.

### 3.5. The Carcinogenic Health Risks of PTEs in Groundwater

The probability distributions of the carcinogenic risk (CR) for two groups (children and adults) under the oral ingestion contact exposure pathways of PTEs in groundwater in the KRB are illustrated in Figure 6. For children, the mean CR value of As was 2.0 × 10^−4^, which indicated a low carcinogenic risk. The CR value of As reached a moderate carcinogenic risk level (1 × 10^−4^) when the cumulative probability reached 60.50%. At the cumulative probability of 96.45%, the CR value of As reached the category of high carcinogenic risk (1 × 10^−3^). Meanwhile, the mean CR value of Cd was 6.9 × 10^−7^, which indicated a very low carcinogenic risk. At the cumulative probability of 85.80%, the CR of Cd reached a low carcinogenic risk level (1 × 10^−6^).

For adults, the mean CR value of As was 3.4 × 10^−5^, which indicated a low carcinogenic risk. At the cumulative probability of 0.12%, the CR value of Cd was lower than the threshold value of carcinogenic risk (1 × 10^−4^). The CR value of As reached the moderate carcinogenic risk level (1 × 10^−4^) when the cumulative probability reached 93.33%. At the cumulative probability of 99.83%, the CR value of As reached the high carcinogenic risk level (1 × 10^−3^). Meanwhile, the mean CR value of Cd was 1.0 × 10^−7^, which indicated a very low carcinogenic risk. At the cumulative probability of 98.70, the CR value of Cd reached the low carcinogenic risk level (1 × 10^−6^).

In terms of the total carcinogenic health risk (TCR) (Figure 7), the TCR values of PTEs in the groundwater in the KRB for adults and children were 3.43 × 10^−5^ and 2.04 × 10^−4^, respectively. The TCR values of PTEs in groundwater for adults and children surpassed the acceptable carcinogenic risk threshold (TCR = 1 × 10^−4^) at a 95% probability. At the cumulative probability of 3.56% and 0.17, the TCR value reached the category of the carcinogenic risk level (1 × 10^−4^). These findings indicate that PTEs in groundwater in the KRB may pose a probabilistic carcinogenic health risk [60]. Moreover, the mean TCR value for children was higher than that for adults, indicating a higher cumulative carcinogenic health risk for children than for adults.

Like the non-carcinogenic health risks, the carcinogenic risks to children of PTEs in groundwater were relatively great, which is related to the higher sensitivity to environmental pollutants of children and their physiological characteristics [31]. In addition, the TCR value of As in the groundwater was higher than that of Cd. This indicates that As poses a higher carcinogenic risk to children. Overall, As is the main pollution element causing carcinogenic health problems in groundwater in the KRB for these two populations. It is worth noting that long-term exposure to As may lead to severe health problems [31,61], of which we should be cautious.

### 3.6. Sensitivity Analysis

A sensitivity analysis was adopted for discussing the influence of each parameter on the health risk assessment results, with larger sensitivity values representing a stronger influence. As shown in Figure 8, all parameters (contents of PTEs, *ET*, and *EF*) showed a positive correlation with the risk assessment results. As shown here, the influence of each parameter on the health risk assessment results showed a similar trend for children and adults. For the non-carcinogenic risks of PTEs in the groundwater, the sensitivity values for children can be ranked as follows: As (55.05%), Se (11.5%), Pb (9.43%), *EF* (9.09%), Mn (8.38%), Al (2.88%), Cd (1.22%), Cu (1.64%), Zn (0.79%), *ET* (0.02%). For adults, the sensitivity values can be ranked as follows: As (54.79%), Se (10.96%), *EF* (10.72%), Pb (9.14%), Mn (7.75%), Al (3.07%), Cu (1.77%), Cd (0.78%), *ET* (0.73%), Zn (0.29%). For the carcinogenic risks of PTEs in the groundwater, the sensitivity values for children can be ranked as follows: As (89.71%), *EF* (8.4%), Cd (1.82%), *ET* (0.07%). For adults, the sensitivity values can be ranked as follows: As (88.62%), *EF* (9.86%), Cd (1.22%), *ET* (0.3%). These results indicate that As in the groundwater in the KRB is the dominant factor influencing the results of health risk assessment.

## 4. Discussion

The traditional health risk assessment model used in this study relied on some deterministic exposure parameters when calculating the CDI values of PTEs in the groundwater [20]. Due to differences in these parameters for specific individuals in different regions, related parameters used in this study might not be suitable for accurately estimating probable health risks [62]. However, this study did more accurately identify the dominant health risk factor in groundwater in the KRB, based on the Monte Carlo Simulation.

Overall, As, among these eight PTEs, is the main pollution factor and health risk factor in groundwater in the KRB. However, As in various water bodies can cause adverse health risks [13]. As was recognized as a Class I carcinogen by the International Agency for Research on Cancer (IARC) of the WHO in 2017. As has the ability to move and transform in different ecosystems, and can threaten the health of entire ecosystems [62]. High-As groundwater is a serious issue worldwide due to its high toxicity and bioaccumulation. Obtained results of the present study emphasized that the probable health risks caused by exposure to As in the groundwater in the KRB cannot be ignored. Therefore, the pollution and probable health risks of As in groundwater in the KRB should have special attention paid to them, considering As’ higher levels and probable health risks in this region.

The exposure parameters (such as *BW* and *SA*) for calculating the CDI used in the present study were obtained from related studies, which might not be very appropriate for analyzing the health risk of PTEs in groundwater in the KRB. Further studies are needed to explore the suitability of exposure parameters to obtain a more accurate estimation of the potential health risks of PTEs in groundwater. Despite these limitations, the present study can clarify our understanding of the pollution risks of PTEs in groundwater in arid zones.

## 5. Conclusions

In conclusion, this study analyzed the pollution and potential health risks of eight PTEs in the groundwater in the KRB for the first time in this region. The Nemerow index (NI) and the health risk assessment model, based on the Monte Carlo Simulation, were adopted for pollution and health risk assessment. Obtained results of this study revealed that the mean contents of Al, As, and Mn in the groundwater surpassed the Class III threshold of the Standard for Groundwater Quality of China (GB/T 14848—2017) by factors of 1.17, 2.10, and 1.80 times, respectively. Results of the pollution assessment revealed that the pollution degree of PTEs in the groundwater decreased in the order of As > Mn > Al > Se > Pb > Zn > Cd > Cu. Among them, As in the groundwater falls into the moderate pollution level, while Al and Mn fall into the low pollution level. Se falls into the slight pollution level, and the other four PTEs fall into the no pollution level. The overall pollution levels of investigated PTEs in the groundwater fall into the moderate pollution level. The spatial distributions of the contents of different PTEs in the groundwater were different, with the distribution of the high-content areas of each toxic element in the groundwater showing a “dotted-distribution” pattern. Results of the health risk assessment revealed that the mean hazard quotient values of PTEs decreased in the order of As > Se > Pb > Mn > Al > Cu > Cd > Zn. Among them, As may pose a non-carcinogenic health risk, whereas the non-carcinogenic health risk posed by the other seven PTEs in groundwater will not have the non-carcinogenic risks. For adults and children, the mean carcinogenic risk index values of As were 3.4 × 10^−5^ and 2.0 × 10^−4^, respectively, at a low carcinogenic risk level. The mean CR value of Cd was 1.0 × 10^−7^ and 6.9 × 10^−7^, respectively, at a very low carcinogenic risk level. Children have a higher cumulative health risk compared to adults. Overall, according to the total risk index, all investigated PTEs in groundwater in the KRB may pose a probabilistic non-carcinogenic health risk and a low carcinogenic health risk. However, As was confirmed as the dominant pollution factor and health risk factor of groundwater in the KRB. The obtained results of this study could serve as valuable tools for groundwater management efforts in arid ecosystems. This information is crucial for identifying the potential risks of PTEs in the future, especially in arid zones with areas suffering from severe water shortages. Such comprehensive analyses are essential for sustainable water management, and can provide the scientific support needed for the prevention and control of PTEs in groundwater in arid zones.

## Figures and Tables

**Figure 1 toxics-12-00474-f001:**
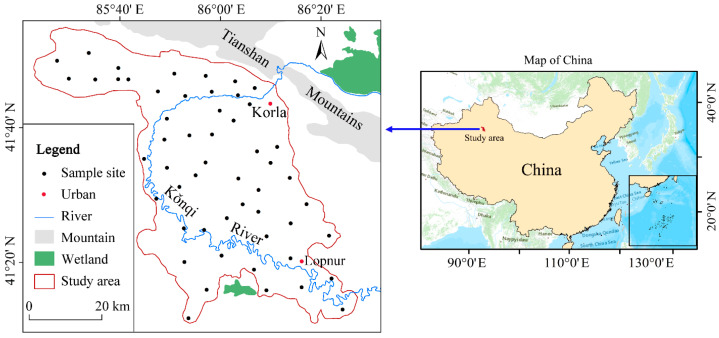
Locations of the Kǒnqi River Basin (KRB) and sample sites.

**Figure 2 toxics-12-00474-f002:**
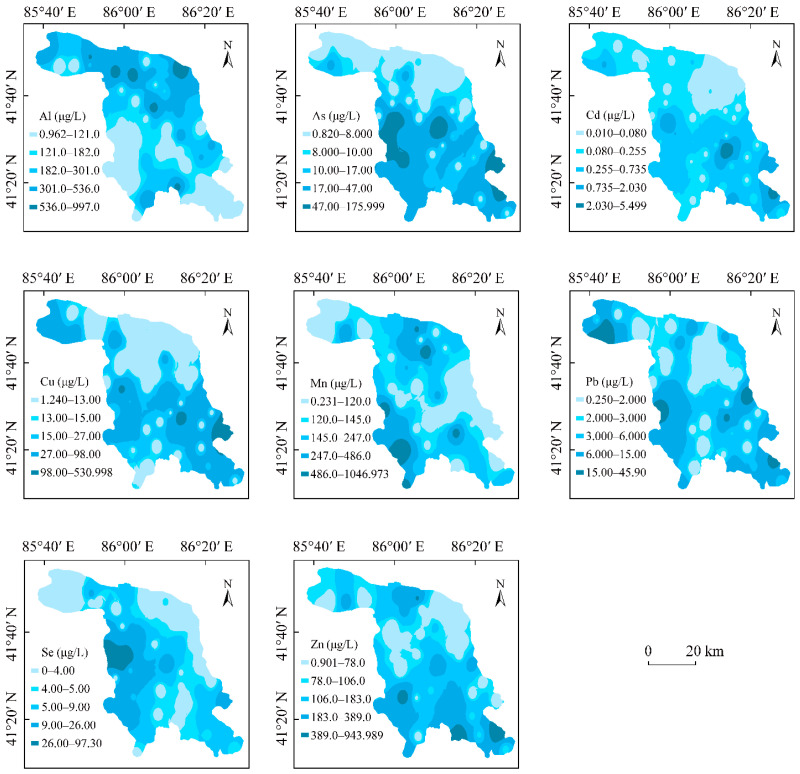
Distribution of PTE content in groundwater in the KRB.

**Figure 3 toxics-12-00474-f003:**
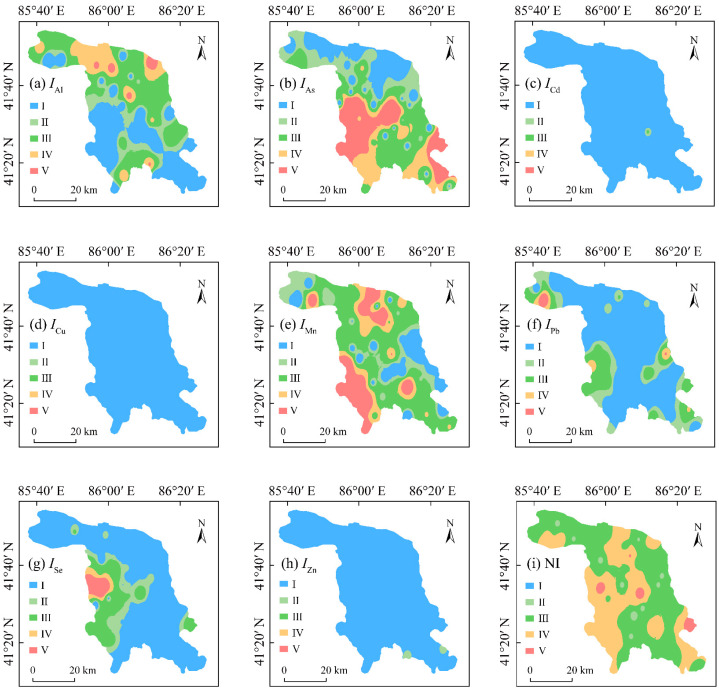
Spatial distribution of pollution levels of PTEs in groundwater in the KRB. (**a**–**h**) represents the pollution level expressed by the single-factor pollution index, while (**i**) denotes the pollution level expressed by the Nemerow pollution index.

**Figure 4 toxics-12-00474-f004:**
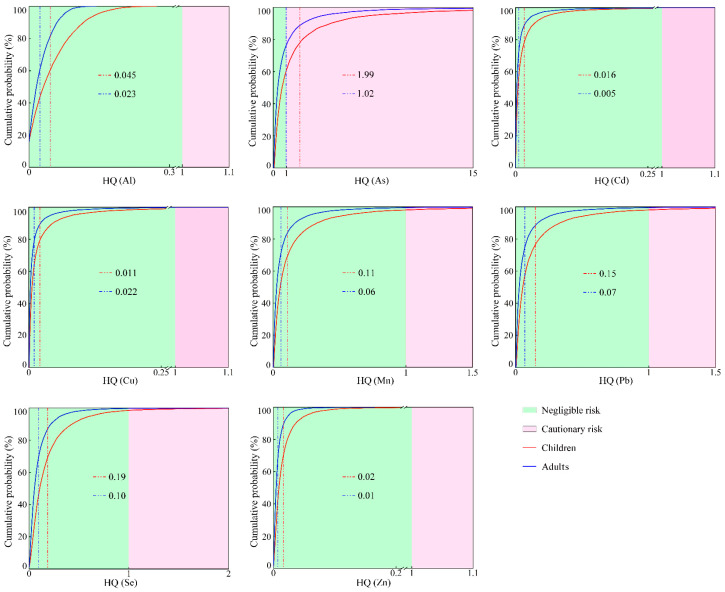
Probability distribution for HQ of PTEs in groundwater in the KRB. (The red and blue vertical dashed lines represent the mean HQ values for children and adults, respectively).

**Figure 5 toxics-12-00474-f005:**
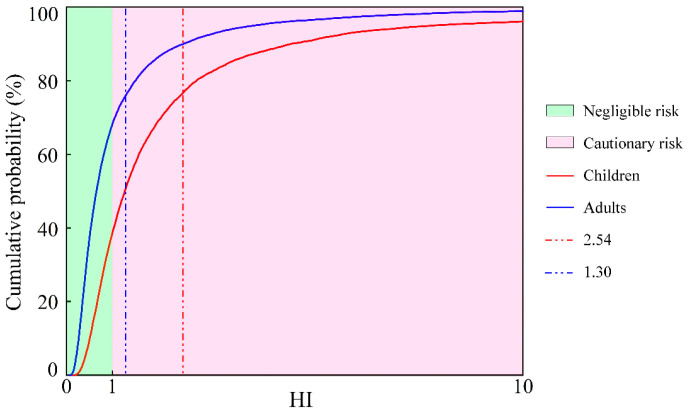
Probability distribution for HI of PTEs in groundwater in the KRB. (The red and blue vertical dashed lines represent the mean HI values for children and adults, respectively).

**Figure 6 toxics-12-00474-f006:**
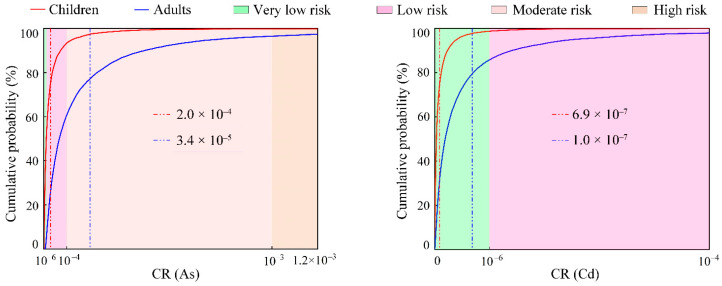
Probability distribution of the CR of HEs in groundwater in the KRB. (The red and blue vertical dashed lines represent the mean CR values for children and adults, respectively).

**Figure 7 toxics-12-00474-f007:**
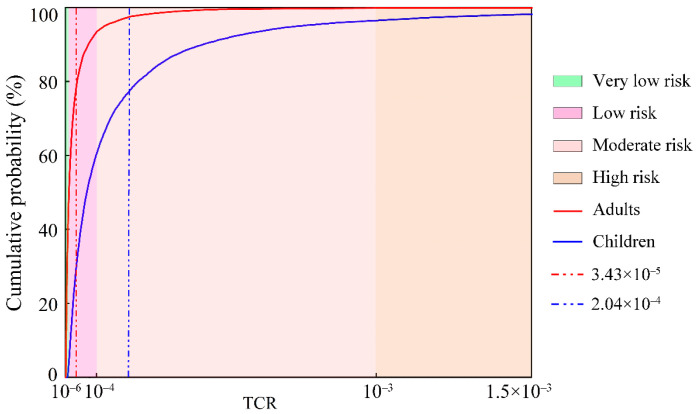
Probability distribution of the TCR of PTEs in groundwater in the KRB. (The red and blue vertical dashed lines represent the mean TCR values for adults and children, respectively).

**Figure 8 toxics-12-00474-f008:**
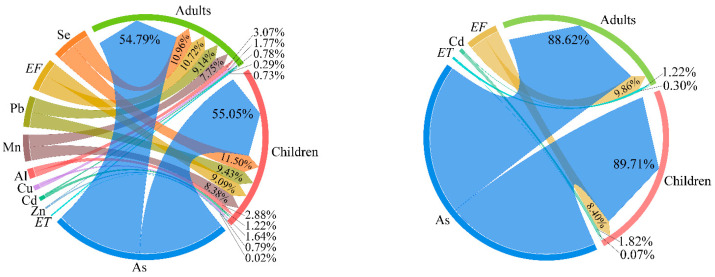
Sensitivity analysis of health risks of PTEs in groundwater in the KRB.

**Table 1 toxics-12-00474-t001:** Classification standards of pollution degree of *I_i_* and NI.

Class	Pollution Degree	*I_i_*	NI
I	No pollution	*I_i_* < 0.7	NI < 0.7
II	Slightly pollution	0.7 < *I_i_* ≤ 1	0.7 < NI ≤ 1
III	Low pollution	1 < *I_i_* ≤ 2	1 < NI ≤ 2
IV	Moderate pollution	2 < *I_i_* ≤ 3	2 < NI ≤ 3
V	High pollution	*I_i_* > 3	NI > 3

**Table 2 toxics-12-00474-t002:** *RfD* and *SF* values for PTEs.

Parameters	Al	As	Cd	Cu	Mn	Pb	Se	Zn
*RfD* _oral_	0.14	0.003	0.0005	0.04	0.046	0.0014	0.001	0.3
*RfD* _dermal_	0.14	0.000285	0.000025	0.012	0.0018	0.00042	0.0015	0.06
*SF* _oral_	/	1.5	6.1	/	/	/	/	/

Note: / indicates no related data.

**Table 3 toxics-12-00474-t003:** Statistical summary of PTEs contents in the groundwater (*n* = 53).

Items	Al	As	Cd	Cu	Mn	Pb	Se	Zn
Minimum/(μg/L)	0.96	0.82	0.01	1.24	0.23	0.25	0	0.90
Maximum/(μg/L)	997.00	176.00	5.50	530.00	1047.00	45.90	97.30	944.00
Mean/(μg/L)	233.00	21.00	0.35	32.00	180.00	6.00	7.00	156.00
Standard deviation/(μg/L)	0.25	0.04	0.00088	0.08	0.23	0.01	0.01	0.19
Coefficient of variation	1.06	1.84	2.52	2.45	1.29	1.37	1.91	1.23
Skewness	1.09	3.13	4.49	5.33	2.06	2.52	5.32	2.61
Kurtosis	0.70	9.45	23.31	32.51	4.16	7.54	32.08	7.78
* National standard/(mg/L)	0.20	0.01	0.005	1.00	0.10	0.01	0.01	1.00
Standard-exceeding ratio/(%)	49.06	35.85	1.87	0	45.28	20.75	15.09	0

Note: * The Class III threshold of the Standard for Groundwater Quality (GB/T 14848—2017).

**Table 4 toxics-12-00474-t004:** Pollution levels of PTEs in the groundwater (*n* = 53).

Items	*I_i_*								NI
	Al	As	Cd	Cu	Mn	Pb	Se	Zn	
Minimum	0.005	0.082	0.002	0.001	0.002	0.025	0	0.001	0.280
Maximum	4.985	17.600	1.100	0.531	10.470	4.590	9.730	0.944	12.600
Mean	1.165	2.107	0.071	0.032	1.804	0.646	0.744	0.156	2.750
CV	1.056	1.844	2.787	2.449	1.289	1.372	1.911	1.228	0.988

## Data Availability

Data will be available upon request to the corresponding author.

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
