# Peer review of "Pollution and Health Risk Assessment of Potentially Toxic Elements in Groundwater in the Kǒnqi River Basin (NW China)"

_toxics, 2024, doi:10.3390/toxics12070474_

Round 1

Reviewer 1 Report

Comments and Suggestions for Authors

Dear Authors

Please see the comments that could improve the document

All the best

Comments on the Quality of English Language

Reviewer 2 Report

Comments and Suggestions for Authors

Dear authors,

The submitted paper "Pollution and Health Risk Assessment of Potentially Toxic Elements in Groundwater in the Kǒnqi River Basin of NW China" fits well within the aim and scope of the journal - Toxics.

Some minor issues should be dealt with before the paper meets the demands for publication in the journal:

1. The MS should follow the structure of the journal's template - the Results and the Discussion should be separated and numbered consequently.

2. The terms should be unified - either use potentially toxic elements (PTEs) or toxic elements (TEs), but not both. Maybe "PTEs" applies better in your case.

 3. Please provide information on whether the laboratory performing the ICP-MS analysis is accredited under ISO/IEC 17025.

4. Paragraph 2.5., Lines 170-171: As and Cd are listed as both non-carcinogenic TEs and carcinogenic TEs.

5. Table 2: Please follow the alphabetical order for listing the PTEs, as done in the other tables.

6. Please provide the DOI numbers for the cited literature as requested in the journal's template.

7. It looks like only one sample was collected from all the 53 points. There is no information on whether the samples were analysed at least in triplicate to obtain meaningful statistics. If so, it should be mentioned and Table 3 should be updated at least with the RSDs or, even better, with the estimation of the uncertainties.

Comments on the Quality of English Language

Dear authors,

The submitted paper "Pollution and Health Risk Assessment of Potentially Toxic Elements in Groundwater in the Kǒnqi River Basin of NW China" fits well within the aim and scope of the journal - Toxics.

Some minor issues should be dealt with before the paper meets the demands for publication in the journal:

1. The MS should follow the structure of the journal's template - the Results and the Discussion should be separated and numbered consequently.

2. The terms should be unified - either use potentially toxic elements (PTEs) or toxic elements (TEs), but not both. Maybe "PTEs" applies better in your case.

 3. Please provide information on whether the laboratory performing the ICP-MS analysis is accredited under ISO/IEC 17025.

4. Paragraph 2.5., Lines 170-171: As and Cd are listed as both non-carcinogenic TEs and carcinogenic TEs.

5. Table 2: Please follow the alphabetical order for listing the PTEs, as done in the other tables.

6. Please provide the DOI numbers for the cited literature as requested in the journal's template.

7. It looks like only one sample was collected from all the 53 points. There is no information on whether the samples were analysed at least in triplicate to obtain meaningful statistics. If so, it should be mentioned and Table 3 should be updated at least with the RSDs or, even better, with the estimation of the uncertainties.

Reviewer 3 Report

Comments and Suggestions for Authors

The article contains interesting and useful information about selected pollutants in groundwater and their potential toxic impact on human health. The Konqi River catchment in China was chosen as an example. In the manuscript, the authors used both the Chinese classification and a number of groundwater quality indices and those related to the impact on human health (adults and children) for 8 parameters (As, Se, Pb, Cu, Al, Zn, Mn, Cd). The introduction, results and discussion of results in the part related to the impact on human health raise no objections. I have a few comments on the remaining parts, i.e. the Discussion of the Results for the spatiotemporal variability of parameters expressed in raw results and converted into water quality indexes, Materials and Methods, and the Conclusions. Overall, the article is interesting and I think that with some minor corrections it will be worthy of publication in Toxics.
- Figure 1: please add scale on all maps.
- Please specify the error of laboratory equipment for individual parameters. - What software was used to perform statistical analyzes and maps?
- Table 2: explain what "/" means.
- Please expand the discussion in subsections 3.1 to 3.3 in the context of possible causes of variability in parameter concentration and comparison of the spatiotemporal variability of these parameters with the results of other authors.
- Please include CV values in the table.
- Why was the indicated interpolation method (Ordinary Kriging Interpolation) used? What makes it stand out? It would be worth describing this issue in Materials and Methods, not in Results and Discussion.
- It would be good to make a classification (ranges) on the maps according to the described local standards in the context of their variability. If the authors believe that the ranges used should remain, I would like to ask them to explain why.
- I would like to ask you to use abbreviations for all elements in the text to maintain consistency.
- Line 297-298: please provide the source for the quoted List.
- In the Conclusions, please additionally include the following issues: a) universality of the results - can they be used for various climatic, hydrological, hydrographic, geological, land use conditions, etc.? b) possible future research directions; c) limitations of the presented research in the context of the proposed methodology; d) possible actions to reduce the negative impact of the tested elements on human health, but also on the environment and the levels of these substances at the source of their occurrence; recommendations to help limit their occurrence/spread (this issue can be included in the Discussion instead of in the Conclusions - at your discretion)

Comments on the Quality of English Language

There are typos, missing spaces in the right places or double spaces. Please check the text for this purpose. For example: line 267-268: no spaces between the element name and the value, line 232: unnecessary "and".
